# Appraisal of Motor Skills in a Sample of Students within the Moldavian Area

**DOI:** 10.3390/bs10060097

**Published:** 2020-06-09

**Authors:** Ionut Onose, Beatrice Aurelia Abalasei, Raluca Mihaela Onose, Adriana Albu

**Affiliations:** 1Faculty of Physical Education and Sports, “Alexandru Ioan Cuza” University of Iasi, 700554 Iasi, Romania; ionut.onose@uaic.ro (I.O.); beatrice.abalasei@uaic.ro (B.A.A.); 2Faculty of Medicine, University of Medicine and Pharmacy “Grigore T. Popa”, 700115 Iasi, Romania; adriana.albu@umfiasi.ro

**Keywords:** bodyweight, motor skills, BMI

## Abstract

The selection of children for training in a certain sports branch should be based on the assessment of their physical development and their motor skills. The aim of the study: the evaluation of the students’ motor skills in relation to Body Mass Index (BMI) in order to orient them towards certain sports branches. Methods: The research study was conducted on a sample of 220 fifth grade students within the Moldavian Area. We have calculated the BMI and we have assessed the motor skills according to the national standards established for each school grade. Results: The BMI values are mainly normal (75.45%), yet there are significant differences in development between students in the three counties under analysis, with the most significant values recorded in the county of Suceava (18.48 ± 0.45 for boys and 18.06 ± 0.48 for girls). As far as the push-ups test grading is concerned, 8.63% of the students achieved below 5; there are also significant differences from one region to another (the highest values were recorded in Iasi 11.05 ± 1.00 for boys, 9.93 ± 0.97 for girls, in Suceava 7.98 ± 0.89 for boys and 4.18 ± 0.46 for girls and in Vrancea 9.97 ± 0.48 for boys and 7.70 ± 0.33 for girls). Softball throw was perfectly executed and graded with 10 by 59.09% of the students. Standing long jump was graded with 10 for only 30.45% of the students. The differences obtained according to *p*-value indicated considerable differences for all motor skills tests and for all study groups. Conclusions: there are substantial differences in children’s physical development and motor skills from one county to another and this aspect is essential in the selection of young people who will practice high performance sports.

## 1. Introduction

In assessing one’s physical development, great attention must be paid to the socioeconomic status of the family, which influences both the family members’ diet and their concern for physical activity [1]. Issues related to a child’s physical development should be approached by analyzing the geographical background of the family, as development-related differences have been observed between children living in urban areas and those living in rural areas [2].

In Romania, for the 1999 standardization at 10 years old, the average value of height for urban males is 137.8 cm and 132.9 cm among rural children, so there is a difference of almost 5 cm between the height of urban and rural students. Among females, the values are 137.2 cm for urban girls and 129.0 cm for rural girls, so there is a difference of over 8 cm [3].

Coaches should be aware of and should consider these differences in physical development related to the child’s geographical background when orienting a student towards a certain sports branch. The selection of a student’s sports branch should be made according to the requirements imposed by the respective activity. In basketball, for instance, the selection process should target taller students, whereas in handball students’ height is less important, and a somatic model is recommended (height between 148 and 152 cm, and weight between 37 and 39 kg for 10-year-olds and height between 154 and 158 cm and weight between 40 and 42 kg for 11-year-olds) [4]. High-demand sports require extraordinary physiological capacities combined with outstanding abilities in the areas of motor control, perception, and cognitive functioning. One recent meta-analysis showed small to medium effects of basic cognitive functions in experts and elite athletes which may point at their superiority in terms of basal cognitive functions [5].

In recent years, a considerable increase of typically sedentary behavior occurring in adults, children and adolescents has been observed. Such behavior seems to be closely connected with the lack of motor experiences and the engagement in physical exercise programs, which may partly impact upon the levels of motor coordination and result in overweight and obesity, particularly in the first years of life [6].

Physical development is assessed with the help of body weight and height parameters that allow the calculation of the body mass index. The values of the body mass index indicate that the young person might be placed within normal values, may be underweight or of overweight (obesity). The prevention of overweight onset is a permanent concern of the specialists in the field of public health. An overweight teenager is very prone to becoming an overweight and then an obese adult [7,8]. 

In addressing the problems related to the physical development of students, the parents’ nutrition-related habits and the systematic practice of physical exercise must also be taken into account. In the specialized literature there are studies in which the correlation is made between the BMI of the parents and that of the child [9,10]. Additionally, there is a correlation between the physical activity submitted by the parents and that submitted by the student. If a student observes his parents paying attention to food intake and physical activity, s/he will be tempted to do the same. In the first years of life, parents act as important role-models, they can provide their children with positive examples that lead to the creation of a healthy lifestyle [11].

Overweight and obese children and adolescents have generally been shown to display lower physical activity levels, poorer motor performance (i.e., motor competence, motor ability, and fundamental movement skills), as well as lower perceived motor competence (self-perception of ability to perform motor skills) when compared to their normal weight peers [12,13]. A plausible explanation for a negative association between BMI and motor coordination level in children and adolescents is based on biomechanical issues. That is, the higher the amount of body mass, the higher the mechanical work required to perform motor tasks, especially those which demand body weight-bearing. [14]. 

In selecting students for high-performance sports, coaches should begin by learning certain aspects related to the physical development and the motor skills of students. Fundamental motor skills (FMS) are important in child development because theoretical frameworks support the idea that FMS form the foundation for future physical activity engagement [15]. Motor skill competence is defined in terms of common fundamental motor skills, specifically, the development of object control (e.g., throwing, kicking) and locomotor skills (e.g., running, jumping, hopping) [16]. 

Among the students practicing systematic physical activity, physical development is important, because among the young females with a normal or asthenic development (malnutrition) there is a better evolution of upper body strength, greater speed and strength of flexor muscles, compared to people with a hypersthenic development (obesity) [17].

## 2. Materials and Methods

### 2.1. Aim of the Study

In this study we have assessed bodyweight and height in the students included in the research sample and used the results obtained by the students in motor skills tests (starting from the national standards). The results obtained have been calculated per county in order to make the correlation between the value of the Body Mass Index and the motor skills of the students.

### 2.2. Participants

The research study was conducted on a sample of 220 fifth graders from the rural areas of the Vrancea County (61 students), Iasi (72 students) and Suceava (87 students) within the Moldavian area. The repartition by gender was unequal, as we have examined 116 boys and 104 girls. In each county, we included within the study the students within two communes, given that the number of students has been on a decreasing trend in the rural areas. The students included in the study are aged between 10 and 12 years old.

### 2.3. Procedure

Subsequently, we made the correlation between the value of Body Mass Index and the motor skills assessed using the aforementioned tests. The tests were applied throughout January-February 2020 to all students, during physical education and sports classes. In order to carry out the tests, we collaborated with the physical education teachers from the respective schools (the performance of these tests is required at the national level).

We have assessed the value of bodyweight and the motor skills of the subjects. Bodyweight was measured using the Body Mass Index. The formula is the same as the one for the adults, but result interpretation differs according to age group and gender [18]. Values between 11% and 84% are normal; values exceeding 84% indicate overweight or obesity, while values under 11% indicate underweight.

The motor skills were assessed by means of 4 tests included in the national grading standard [17] for the fifth grade: push-ups; standing softball throw; standing long jump; upper body extension, face down. The grade standard differs according to gender. To avoid a significant dispersion of the results, the assessment starts from the grades 5 and 6–7, 8 and 9–10 (Table 1).

### 2.4. Statistical Analysis

The statistical analysis applied in this paper aimed at identifying and quantifying the links between body mass index (BMI) and a number of potentially explanatory, quantitative variables (number of push-ups, softball throwing in m, standing long jump in m and number of upper body extensions). The statistical calculations as well as the graphical representations were made using Excel 2016.

For this purpose, the linear regression method was applied, which is a statistical method that allows the identification and quantification of the links caused between one (single regression) or more (multiple regression) quantitative explanatory variables (predictors) and a dependent variable (response), expressed, also quantitatively.

Regarding the values of the coefficient, they can oscillate between −1 and 1, indicating the intensity of the relationship between the variables; therefore, the closer the value is to −1 or 1, the closer the relationship.

Negative values indicate inverse proportionality relationships while positive values indicate direct proportionality relationships. Raised to square, the correlation coefficient becomes the coefficient of determination (R^2^), expressing the fraction of the variance of the dependent variable explained by the explanatory variables.

In addition, the validation of the regression model involves testing the statistical significance of the correction coefficient, the free coefficient and the partial regression coefficient. For this purpose, a series of statistical tests are used, for which the probability of error must be less than 5% (*p* < 0.05).

### 2.5. Ethics Statement

All subjects gave their informed consent for inclusion before they participated in the study. The study was conducted in accordance with the Declaration of Helsinki, and the protocol was approved by the Ethics Committee of the Faculty of Physical Education and Sports of “Alexandru Ioan Cuza” University of Iasi, Romania (No 128 / 29.01.2020).

## 3. Results

The study focuses on two directions, namely the bodyweight of students and their motor skills. Bodyweight ranges mostly within normal limits (75.45%). (Table 2).

Regarding the BMI, the highest values in the case of boys were found in the Suceava region, with an average of 18.42 ± 0.45, variation limits between 14.43 and 29.70. The studied character displayed a good homogeneity, the value of the coefficient of variation being 15.87% (Table 3).

With regard to the male subjects from the three regions, the lowest values were recorded in the Iasi region, with an average range of 17.68 ± 0.53, a minimum value of 13.68 and a maximum value of 30.40; these values led to a coefficient of variation of 19.32% (Table 3)

With regard to female subjects, the highest values were recorded in the Suceava region, with an average range of 18.06 ± 0.48. The lowest values were recorded in the Vrancea region, where the average was 16.07 ± 0.42.

Table 4 shows us the number of students that performed the motor skills and what grade did they obtained according to the national standards (see Table 1).

Table 5 highlights the results on the correlation between body mass index (BMI) and the motor skills tests. 

As Table 5 indicates, the average value for BMI of the boys in the Iasi region was 17.68 ± 0.53, and the average in terms of the number of push-ups was 11.05 ± 1.00 reps.

Figure 1a shows the relationship between BMI and the number of push-ups performed by boys, and a positive regression can be observed (F6.00 > Fsig. 0.018) (F.sig. is the threshold value over which the results of the analysis significantly defines), therefore, the ability to perform push-ups is influenced in 53% by the BMI for the boys in Iasi region (Figure 1a).

With regard to the male subjects from the Suceava region, the average value for BMI was 18.48 ± 0.45 and the average number of push-ups was 7.98 ± 0.89 repetitions (Table 5). Regarding the influence of the ability to perform push-ups by BMI, the value of the coefficient of determination (R^2^) is 5 × 10^−6^, the regression being also positive (F0.88 > Fsig. 0.35) (Figure 1b).

For the Vrancea region, the average values obtained in terms of BMI were 18.32 ± 0.41, while for the number of push-ups the average was at a level of 9.97 ± 0.48 reps. In this case, a positive regression was recorded (F0.59 > Fsig. 0.44), therefore the ability to perform push-ups is influenced in 19% of the cases by the BMI (Figure 1c).

From a statistical point of view, significant differences were highlighted, the values of *p*-value being higher than 0.001 (Table 5).

With regard to the female subjects from the three analyzed regions we can observe, similarly as in the case of the male subjects, that there is a positive regression. Therefore, for girls in the Iasi region, the mean BMI was 16.55 ± 0.58 and the number of push-ups was 9.39 ± 0.97 reps (Table 5). Regarding the significance of the differences between the averages, the difference is very significant (*p*-value = 8.99 × 10^−15^), this being lower than the value 0.001. Regression analysis F2.21 > Fsig. 0.14 shows a positive regression, therefore, the ability to perform push-ups is influenced in 17% of the cases by the BMI (Figure 2a).

The analysis of BMI data correlated with the number of push-ups performed by the girls in the Suceava region shows a positive regression (F2.82 > Fsig. 0.10), the ability to perform push-ups being influenced by only 0.1% (Figure 2b).

For the girls from the Vrancea region the average BMI was 16.07 ± 0.42 and for the number of push-ups an average of 7.70 ± 0.33 reps was obtained; *p*-value was 9.93 × 10^−10^ the difference being also very significant (*p*-value > 0.001) (Table 5). The ability to perform push-ups is influenced in 15% girls by the BMI (R^2^ = 0.15) (Figure 2c).

Another test analyzed was the softball throw; the average values obtained for both boys and girls also correlated with the BMI.

Therefore, for the boys from the Iași region, the average value obtained for this exercise was 27.27 ± 0.83 m, the coefficient of determination (R^2^) being equal to one (Figure 3a) with the regression being a negative one (F0,18 > Fsig. 0.66), therefore, the BMI does not influence the results obtained in this test.

The boys from the Suceava region recorded an average of 24.12 ± 0.93 m (Table 5), the correlation in this case being a positive one (F1.48 > Fsig. 0.22), indicating that the softball throwing test was influenced by the BMI (R^2^ = 0.035) (Figure 3b).

For the boys from the Vrancea region, as well as for those from Iași, the regression was negative (F0.37 > Fsig. 0.54), the average for this sample being 21.23 ± 0.57 m. (Table 5). The difference regarding the *p*-value (9.73 × 10^−8^) is highly significant (Figure 3c).

For the girls from the Iași region, the regression was positive, R^2^ = 0.0012 (Figure 4a). For girls, the correlation between the softball throwing test and BMI showed a negative regression for the group from the Suceava region (F0.54 > Fsig. 0.46), R^2^ recording a value of 0.0127 (Figure 4b) and for the one from the Vrancea region where the value of R^2^ was equal to one (Figure 4c), F3.89 > Fsig.0.058.. With regard to the *p*-value, the data displayed in Table 5 indicates that the differences are highly significant.

The standing long jump correlated with BMI for the groups of boys from the Iași and Suceava regions; the regression was positive F1.13 > Fsig. 0.29 for the Iași group and F4.45 > Fsig. 0.041 for the Suceava group. Negative regression F0.078 < Fsig 0.781 was recorded in boys from the Vrancea region (Figure 5a–c).

For the girls from the three analyzed regions, the averages obtained for the softball throwing test were 1.34 ± 0.44 m for the Iași region, 1.40 ± 0.03 m for the Suceava region and 1.33 ± 0.03 m for the Vrancea region (Table 5). The regression for the groups of girls from Iași and Suceava was positive: F1.69 > Fsig. 0.20 and F18.02 > Fsig. 0.0001 (Figure 6a,b). The values obtained for the *p*-value were 1.03 × 10^−5^ for the girls from the Iași region and 4.79 × 10^−12^ for those from the Suceava region.

For the group of girls from the Vrancea region, a negative regression (F0.15 < Fsig 0.69) was obtained, the differences being also quite significant, *p*-value = 8.12 × 10^−5^. The values obtained for the coefficient of determination (R^2^) were 1 (Iasi), 0.30 (Suceava) and 1 (Vrancea) (Figure 6a–c).

For the Trunk Extensions Test, the boys from the Iași region recorded an average of 33.17 ± 2.03 reps (Table 5). The correlation between BMI and this type of test, in this case, was a positive one— F0.65 > Fsig. 0.42, R^2^ = 1 (Figure 7a).

For those in the Suceava region, the recorded average was 25.37 ± 0.74 reps and the correlation between BMI and this test was also positive—F6.60 > Fsig. 0.01, R^2^ = 1 (Figure 7b).

The last group analyzed, the one in the Vrancea region, recorded an average of 30.94 ± 1.74 and the correlation between BMI and this test was negative (F0.23 < Fsig. 0.63) with an R^2^ = 1 (Figure 7c).

Regarding the female group from the Iasi region, the recorded average was 31.90 ± 1.62 reps (Table 5). The analysis of the correlation between BMI and the upper body extension test was positive—F1.99 > Fsig. 0.17, R^2^ = 1 (Figure 8a).

The group of girls from the Suceava region recorded an average of 23.36 ± 0.69 reps and the correlation between BMI and this test is F5.68 > Fsig. 0.002, the regression being positive, with R^2^ = 1 (Figure 8b).

The representative female group from the Vrancea region had an average of 24.23 ± 1.22 reps and the regression analysis showed a positive correlation with F6.96 > Fsig. 0.001 with R^2^ = 1 (Figure 8c).

## 4. Discussion

Numerous studies show that the environment can influence the physical activity of an individual and indirectly impact upon his/her health [19]. Populations living at high altitudes have a slight delay in linear growth and a larger diameter and chest circumference than the inhabitants of regions closer to the sea level. These differences are generally attributed to the phenomenon of hypobaria, as well as to various socioeconomic, nutritional and environmental factors [20].

The human body prioritizes which segments to grow when there is limited energy available for growth, as it happens at high altitudes. This comes at the expense of other segments, for example the lower arm. The body may prioritize full growth of the hand because it is essential for manual dexterity, whilst the length of the upper arm is particularly important for strength [21].

The present study analyzed the results obtained by students working in schools located on different altitudes (200–250 m altitude for Iasi, 300–350 m altitude for Vrancea, 500–550 m altitude for Suceava).

The study was oriented towards two main directions represented by the assessment of the anthropometric indicators and of the physical activity submitted by the students from the study group. The anthropometric indicators we used were height and body weight that allowed the calculation of the body mass index [22]. The results obtained a range from underweight to overweight. In a study conducted on 11-year-olds in Macao, 17.8% of the male students were underweight, 49.0% had normal weight, 16.2% overweight. As far as the female students were concerned, 16.5% were underweight, 61.7% had normal weight, 11.7% were overweight and 10.1% obese. In the above-mentioned study, obesity and overweight were higher among boys than among girls [23]. Our study indicates differences according to gender, as normal values are featured by 86.20% of the boys and 63.46% of the girls, while overweight–obesity was recorded in 10.34% of the boys and 9.61% of the girls. Another study, conducted on adolescents in Korea, indicated, among males, 3.3% underweight cases, 50.8% with normal values and 46.0% overweight or obese. Among females, the same study found 15.7% underweight students, 59.4% with normal values and only 24.9% overweight or obese [24]. A study conducted on children aged between 11 and 14 in Austria reports the presence of insignificant differences between the prevalence of overweight/obesity among girls (17.1%) and boys (23.2%) [25]. 

There are significant differences in BMI values in the African population, which indicates that local specificity must be taken into account. Normal values are displayed by 49.4% of the students in Djibouti and by 71.3% in Benin. Overweight classifications were detected in 8.7% of the young people in Ghana and reached 31.4% in Egypt. At the same time, values indicating underweight ranged from 12.6% in Egypt to 31.9% in Djbouti. Public health professionals in each country should be aware of these results and should carefully consider them [26].

In adolescents in Iran, there are differences in the value of BMI even between different regions of the country. The average value of BMI varies from 17.66 in the Western part of the country to 19.17 in the North-North-Eastern area. Additionally, the prevalence of overweight cases ranges from 6.29% in the Western area to 13.91% in the North-Northeastern area with statistically significant differences [27].

The second aspect analyzed in the study is related to the physical activity and the results obtained by the students according to the national grading scales. It is important to know the motor skills of students in order to guide their orientation towards certain sports. In a selection for handball, a motor model for 10-year-old students includes softball throw at 23–25 m, standing long jump at 1.70–1.75 m., explosive power at 25–27 cm. For 11-year-olds, softball throw requirements are 25–27 m., for standing long jump 1.75–1.80 m, while for explosive power the requirement is 28–30 cm [4]. 

The percentage of students rejected by coaches during the selections reaches 20%. Students are rejected because their motor skills are not adapted to the specificity of sports (34%), due to behavioral problems (36%) or anthropometric issues (21%) [28].

In the Republic of Moldova, the school curriculum provides 2 h of physical education per week. Of the rural students, 89.8% participate effectively in these classes, with only 4.1% of students exempted for medical reasons. Most (33.1%) perform high intensity physical activities during the lesson for 10–20 min. Furthermore, 24.1% perform such activities in less than 10 min, whereas 29.0% perform them in over 30 min [29].

Push-ups were executed flawlessly (grade 10 level) by 37.27% of the students within our study sample, while in another research carried out in the Suceava County, 42.4% of the students executed the test perfectly (graded with a 10) [30] (Table 4). The Austria study shows 15 push-ups in 40 s, with significant differences according to gender [25].

In softball throw, 3.63% of the cases were below the minimal grade, while perfect executions (graded with 10) were performed in 59.09% of the cases. In a study conducted on urban students, 12.19% were graded below 5 and 57.31% were graded with a 10 [31] (Table 4).

The Korean study indicates that 15.6% of the boys and 31.8% of the girls conduct minimum physical activity [23]. Among the adolescents in Austria, no differences were found between boys (46.2%) and girls (51.4%) as far as exercising was concerned. Boys scored better in 20-m sprint, while girls had better results in balance standing and 6-min run [24].

In standing long jump, the situation is rather alarming, as 46.81% of the students failed to pass, while the study conducted on urban students found only 23.78% executions that were graded below 5 [23] (Table 4). These results are important for coaches within various sports branches, who should take into account complex standards, adapted to the requirements of each sports branch. Furthermore, psychomotricity should be tested, because the body scheme elements, laterality, spatial and temporal orientation are essential in sports activities [32].

The participation of young people in physical activities favors the development of motor skills and has a major contribution to good health. From a psychological perspective, they favor leadership skills, self-discipline, as well as respect for authority, competitiveness and cooperation [33]. 

Most studies focused on health aspects take into account the physical activity performed especially by young people. A study conducted on teenagers in the USA in 1994–1995 and 1996 focusses on these very aspects. Overall, 51.34% of the responses provided by male participants indicate that they exercised three or more times in the past week, while 48.66% of the participants indicated exercising twice a week or less. The results recorded in 1996 did not indicate major changes: 51.21% of the young people performed physical activity at least three times a week and 48.79% less than two times. The results are also similar in female respondents: in 1994–1995, 52.27% of the girls indicated practicing physical activities three or more times a week and 47.73% only twice a week or less. The situation remains fairly similar in 1996 with 53.12% of the female respondents performing physical exercise three or more times a week and 46.88% exercising less than that [34].

Pilot studies on obese patients show improved motor skills in students included in chronic disease control programs. In a study conducted in Italy on students aged between 10 and 12 years, there is an obvious improvement in motor skills after 6 months of treatment. In males, when jumping vertically, there is an evolution from 19.58 ± 6.28 cm to 22.57 ± 8.47 cm, and when throwing the medicine ball, an increase in distance from 4.08 ± 0.65 m to 4.50 ± 0.88 m. Similar positive results are recorded in female respondents. In the vertical jump, there is an increase from 15.78 ± 8.17 cm to 17.83 ± 4.39 cm, whereas in the throw there is an increase from 3.53 ± 0.65 m to 4.07 ± 0.67 m [35].

As far as the limits of our study are concerned, the following aspects should be mentioned: the study was conducted on a limited sample of students in Romania. Moreover, it included students from just one region in Romania. It is necessary to carry out such studies on a significant statistical sample at a national level. We should also add that is very important to take into account the mental and cognitive characteristics of the students when orienting them towards high-performance sports. 

## 5. Conclusions

Our study had two main directions, namely the assessment of bodyweight and of the motor skills and the relationship between these two elements. Body Mass Index values were mostly normal, which is encouraging for the students we have examined. 

Statistical calculations indicate significant differences in BMI values in boys from different counties of Moldova. The same differences occur in girls. The highest values are displayed in both boys and girls in the county of Suceava. There are also considerable differences between the development of boys and girls, an aspect which specialists should be aware of and take into account.

Motor skills were assessed using four tests included in the national grading standard for the fifth grade. In all tests, there were students who scored below grade 5. The percentage of students who received the maximum grade differs from one test to another, an aspect which should be considered by coaches recruiting students for different sports branches. 

For the push-ups test, the best results were recorded in the county of Iasi and the poorest in the county of Suceava. The poorest results in the softball throwing test were recorded in the county of Vrancea. In the standing long jump test, the lowest values for boys were recorded in the county of Iasi and the poorest results for girls in the county of Vrancea. At the trunk extension, the weakest results were recorded in the county of Suceava. These results are of significant importance for coaches from different sports fields who can thus focus on certain geographical areas to select young people with the proper motor skills for a particular sport. 

The BMI push-ups test showed a positive regression for all the groups under analysis, both in the tests performed by boys and the ones performed by girls.

In the softball throwing test for boys, the regression was negative in the counties of Iasi and Vrancea (there was no correlation between BMI and the softball throwing test) as opposed to the positive regression recorded in Suceava.

For the group of girls from the Suceava and Vrancea counties, the correlation between BMI and softball throwing test was negative, whereas for the girls from Iasi, the correlation between BMI and softball throwing was positive.

In the standing long jump test for the groups of boys from Iasi and Suceava, the regression analysis was positive, while the analysis of the correlation between BMI and this test in boys from the Vrancea region, regression was negative. In the groups of girls from the Iasi and Suceava regions, the comparison between BMI and the applied test showed a positive regression, while in the girls from Vrancea, the resulting regression was negative.

In the trunk extension test, the regression for the boys was positive in the Iasi and Suceava regions and negative in the Vrancea region compared to the groups of girls who, when analyzing the correlation between BMI and the applied test, showed a positive result.

The differences obtained according to *p*-value showed very significant differences for all motor skills tests and for all study groups.

Motor skills represent just one of the essential elements in the orientation towards a particular sport. It is equally important to assess one’s physical development while being well aware of one’s mental characteristics, especially those related to perception and decision-making.

Such studies are important for coaches and for all the specialists interested in the normal evolution of students.

## Figures and Tables

**Figure 1 behavsci-10-00097-f001:**
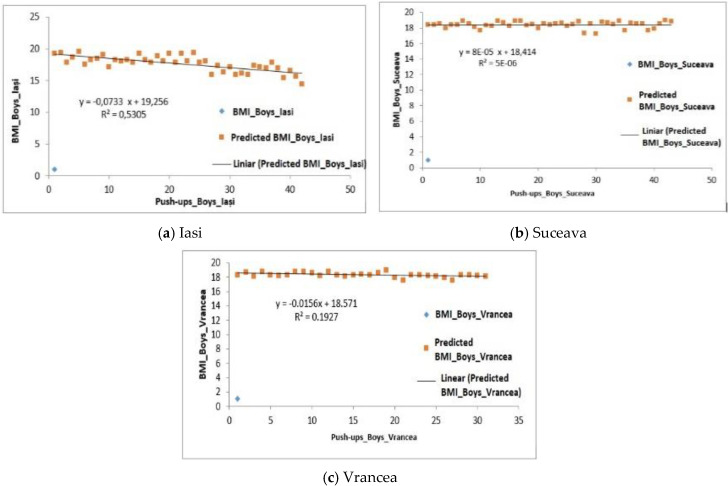
Relationship between the BMI and the number of push-ups performed by the male subjects in the three regions.

**Figure 2 behavsci-10-00097-f002:**
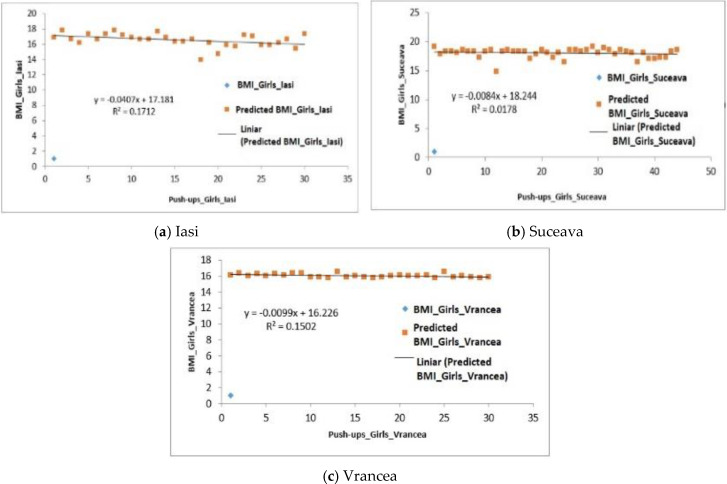
Relationship between the BMI and the number of push-ups performed by the girls in the three regions.

**Figure 3 behavsci-10-00097-f003:**
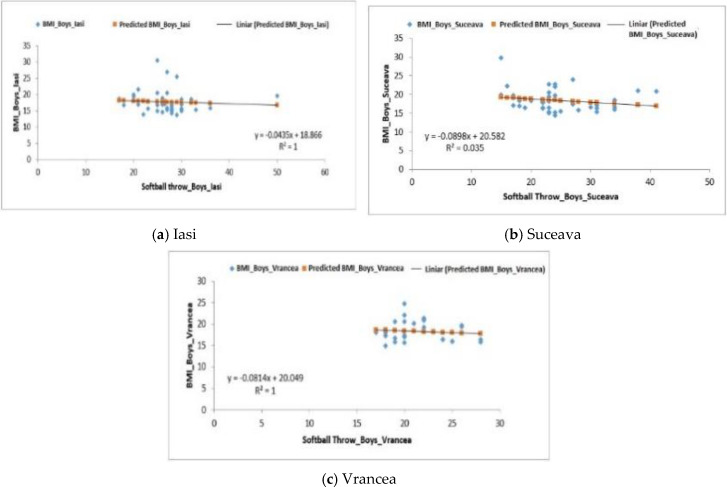
Relationship between the BMI and the softball throwing test for the male subjects in the three regions.

**Figure 4 behavsci-10-00097-f004:**
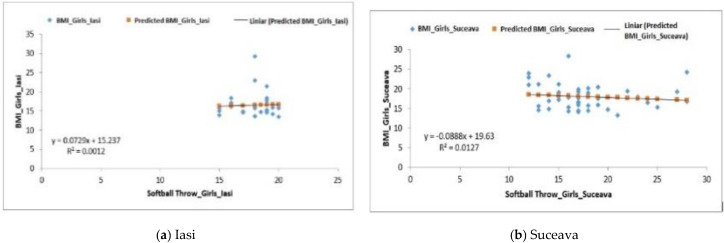
Relationship between the BMI and the softball throwing test for the female subjects in the three regions.

**Figure 5 behavsci-10-00097-f005:**
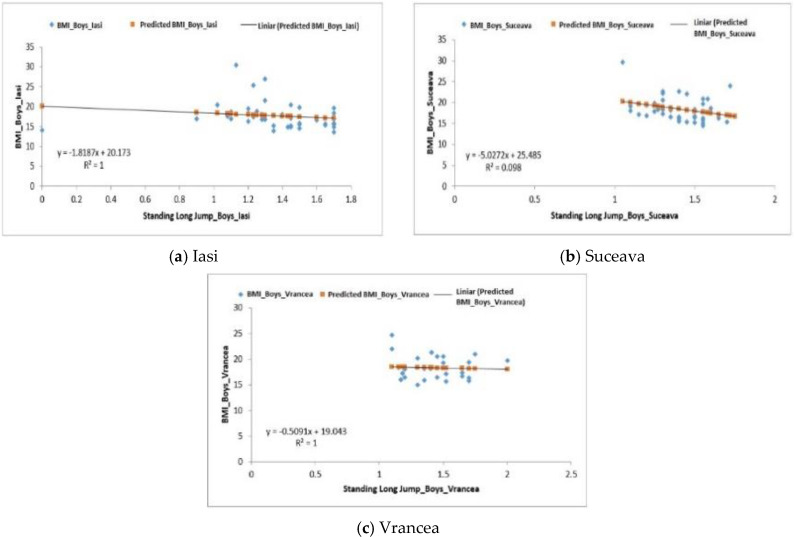
Relationship between the BMI and the standing long jump test for the boys in the three regions.

**Figure 6 behavsci-10-00097-f006:**
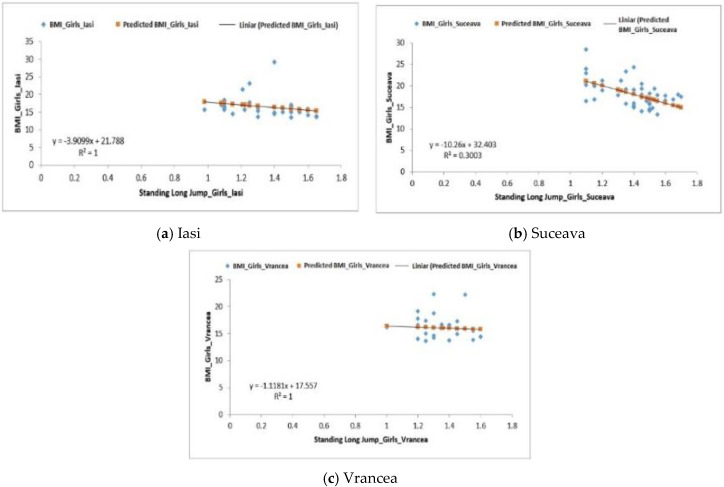
Relationship between the BMI and the standing long jump test for the female subjects in the three regions.

**Figure 7 behavsci-10-00097-f007:**
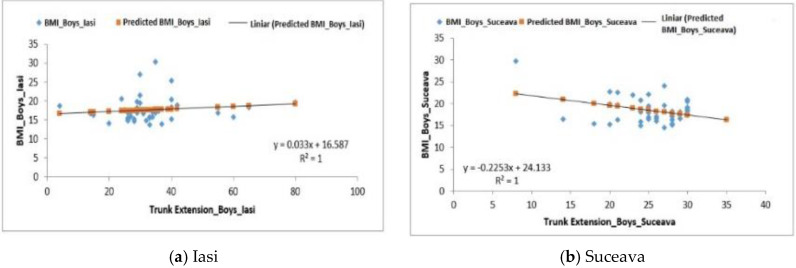
Relationship between the BMI and the upper body extension test for the boys in the three regions.

**Figure 8 behavsci-10-00097-f008:**
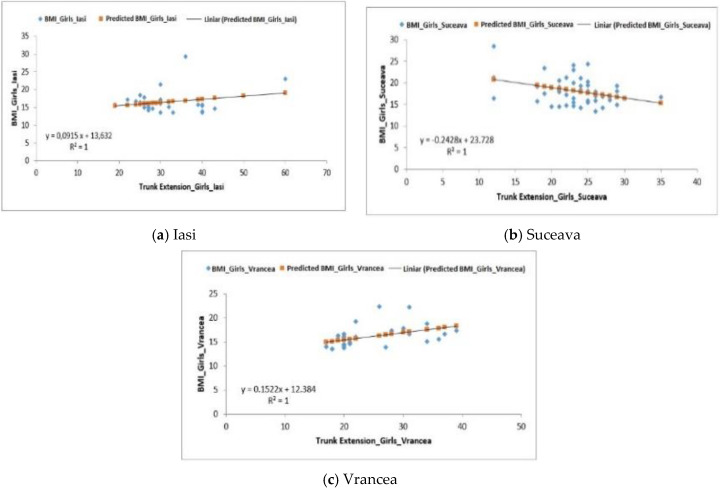
Relationship between the BMI and the upper body extension test for the female subjects in the three regions.

**Table 1 behavsci-10-00097-t001:** National grade standard in the fifth grade.

Vote	Fifth Grade			
	Under 5	5–6	7–9	10
Push-ups (rep.) M	Under 4	4–6	8–12	14
Push-ups (rep.) F	Under 2	2–3	4–6	7
Standing softball throw (m) M	Under 18	18–19	20–22	23
Standing softball throw (m) F	Under 12	12–13	14–16	17
Standing long jump (cm.) M	Under 150	150–153	156–162	165
Standing long jump (cm.) F	Under 130	130–133	136–142	145
Face down, upper body extension (rep.) M	Under 16	16–17	19–23	25
Face down, upper body extension (rep.) F	Under 15	15–16	17–21	23

**Table 2 behavsci-10-00097-t002:** Body Mass Index (BMI) values among the research students.

BMI Values	Underweight	Normal	Overweight	Total
	Evaluation Per County			
Vrancea	10	47	4	61
Iasi	12	54	6	72
Suceava	10	65	12	87
Total number	32	166	22	220
%	14.54	75.45	10.00	
	**Results by the Gender of Students**			
Male	4	100	12	116
Female	28	66	10	104

**Table 3 behavsci-10-00097-t003:** Distribution of results according to gender and county.

COUNTY	SEX	*N*	$\bar{X}\pm s_{\bar{X}}$	V%	MIN	MAX
**IAșI**	M	42	17.68 ± 0.53	19.32	13.68	30.40
	F	30	16.55 ± 0.58	19.33	13.51	29.24
**SUCEAVA**	M	43	18.42 ± 0.45	15.87	14.43	29.70
	F	44	18.06 ± 0.48	17.80	13.31	28.44
**VRANCEA**	M	31	18.32 ± 0.41	12.45	14.95	24.69
	F	30	16.07 ± 0.42	14.15	13.60	22.30

**Table 4 behavsci-10-00097-t004:** The results in motor skills tests.

Test results	Under 5	5–6	7–9	10
	Push-ups			
Total no.	19	48	71	82
%	8.63	21.81	32.27	37.27
	Standing softball throw			
Total no.	8	26	56	130
%	3.63	11.81	25.45	59.09
	Standing long jump			
Total no.	103	33	17	67
%	46.81	15.00	7.72	30.45
	Face down, upper body extension			
Total no.	8	3	56	153
%	3.63	1.36	25.45	69.54

**Table 5 behavsci-10-00097-t005:** The correlation between BMI and the motor tests applied by regions.

MST	Sex	Iasi				Suceava				Vrancea			
		*n*	BMI	MTR	*p*-Value	*n*	BMI	MTR	*p*-Value	*n*	BMI	MTR	*p*-Value
			$\bar{X}\pm s_{\bar{X}}$				$\bar{X}\pm s_{\bar{X}}$				$\bar{X}\pm s_{\bar{X}}$		
**Push-ups**	M	42	17.68 ± 0.53	11.05 ± 1.00	2.27 × 10^−22 ***^	43	18.42 ± 0.45	7.98 ± 0.89	2.24 × 10^−26 ***^	31	18.32 ± 0.41	9.97 ± 0.48	1.56 × 10^−11 ***^
	F	30	16.55 ± 0.58	9.93 ± 0.97	8.99 × 10^−15 ***^	44	18.06 ± 0.48	4.18 ± 0.46	8.45 × 10^−26 ***^	30	16.07 ± 0.42	7.70 ± 0.33	9.39 × 10^−10 ***^
**Softball throw**	M	42	17.68 ± 0.53	27.27 ± 0.83	4.23 × 10^−8 ***^	43	18.48 ± 0.45	24.12 ± 0.93	4.20 × 10^−14 ***^	31	18.32 ± 0.41	21.23 ± 0.57	9.73 × 10^−8 ***^
	F	30	16.55 ± 0.58	18.02 ± 0.28	0.039 ***	44	18.06 ± 0.48	17.73 ± 0.62	2.81 × 10^−11 ***^	30	16.07 ± 0.42	15.67 ± 0.46	0.0001 ***
**Standing long jump**	M	42	17.68 ± 0.53	1.37 ± 0.05	2.22 × 10^−10 ***^	43	18.48 ± 0.45	1.41 ± 0.03	2.83 × 10^−9 ***^	31	18.32 ± 0.41	1.42 ± 0.04	4.84 × 10^−8 ***^
	F	30	16.55 ± 0.58	1.34 ± 0.04	1.04 × 10^−5 ***^	44	18.06 ± 0.48	1.40 ± 0.03	4.79 × 10^−12 ***^	30	16.07 ± 0.42	1.33 ± 0.03	8.12 × 10^−5 ***^
**Upper body extensions**	M	42	17.68 ± 0.53	33.17 ± 2.03	3.64 × 10^−14 ***^	43	18.48 ± 0.45	25.37 ± 0.74	2.16 × 10^−13 ***^	31	18.32 ± 0.41	30.94 ± 1.74	5.44 × 10^−14 ***^
	F	30	16.55 ± 0.58	31.90 ± 1.62	9.71 × 10^−7 ***^	44	18.06 ± 0.48	23.36 ± 0.69	2.11 × 10^−12 ***^	30	16.07 ± 0.42	24.23 ± 1.22	2.71 × 10^−9 ***^

MST-Motor skills tests; MTR = Mean of test results (rep./m); M = male; F = female; $\bar{X}\pm s_{\bar{X}}$ = mean and standard deviation; *** *p* < 0.001 = very significant.

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
