# Peer review of "Appraisal of Motor Skills in a Sample of Students within the Moldavian Area"

_behavsci, 2020, doi:10.3390/bs10060097_

Round 1

Reviewer 1 Report

In this manuscript, the authors examined the relationship between BMI and the motor skills. Toward this end, the authors used data from 220 students of the 5th grade, within the Moldavian Area. Although it is an interesting manuscript, I recommend authors taking into account some modest recommendations that could improve the paper:

Introduction

Line 26-47: The paragraph contains different ideas. I suggest authors to restructure the paragraph, into smaller paragraphs, putting the ideas together, because it is confusing. Line 59-62: Remove the detailed information regarding handball, as it is not related to the study.

In general, restructure the introduction, as it is confusing and does not gather ideas.Methods

- Has it been evaluated whether the students previously performed any sport? How might this influence the results?

- In this section the sections “Procedure” and “Data Analysis” are missing.

- What statistical program did the authors use?

Results

  • Being a descriptive study of a very specific population, it is difficult to extrapolate the results. Therefore, I suggest that the authors carry out a more demanding analysis such as a Regression analysis.
  • Why didn't the authors do a T or Anova test between genders.
  • Being a descriptive analysis, I consider it insufficient. The authors could make a difference by gender and by area.
  • I suggest dividing the "Resuls" section into three parts. 1. Descriptive analysis; 2. Regression analysis; 3. Differential analysis. For a better understanding of the reader.
  • It appears that there are differences by gender, but what analysis have the authors done? It is not clear.

Discussion

- The discussion is poor, and does not relate the results to enough recent studies.

- There are no practical applications of the study or limitations. What was intended with this study? What does it respond to? What practical implications does this study have? What can be improved for subsequent similar studies?

Although it is an interesting study, I consider that the authors should further exploit the data obtained, and restructure introduction and discussion, focusing more on the study objective. Very important to describe the importance of the study and practical implications.

Author Response

Top of Form

Open Review

English language and style

( ) Extensive editing of English language and style required 
( ) Moderate English changes required 
( ) English language and style are fine/minor spell check required 
(x) I don't feel qualified to judge about the English language and style 

Yes

Can be improved

Must be improved

Not applicable

Does the introduction provide sufficient background and include all relevant references?

( )

( )

(x)

( )

Is the research design appropriate?

( )

( )

(x)

( )

Are the methods adequately described?

( )

(x)

( )

( )

Are the results clearly presented?

( )

(x)

( )

( )

Are the conclusions supported by the results?

( )

(x)

( )

( )

Comments and Suggestions for Authors

In this manuscript, the authors examined the relationship between BMI and the motor skills. Toward this end, the authors used data from 220 students of the 5th grade, within the Moldavian Area. Although it is an interesting manuscript, I recommend authors taking into account some modest recommendations that could improve the paper:

Introduction

Line 26-47: The paragraph contains different ideas. I suggest authors to restructure the paragraph, into smaller paragraphs, putting the ideas together, because it is confusing.

We did the modifications line 29-43

Line 59-62: Remove the detailed information regarding handball, as it is not related to the study.

We did the modifications .

In general, restructure the introduction, as it is confusing and does not gather ideas.

We have restructured the introduction.

Methods

- Has it been evaluated whether the students previously performed any sport? How might this influence the results?

The students didn’t performed sports…but it can be a good way to make the selection in sports.

- In this section the sections “-Procedure” and “Data Analysis” are missing.

We did the modifications

- What statistical program did the authors use?

We redid the data analisys through Anova.

Results

  • Being a descriptive study of a very specific population, it is difficult to extrapolate the results. Therefore, I suggest that the authors carry out a more demanding analysis such as a Regression analysis.
  • Why didn't the authors do a T or Anova test between genders.
  • Being a descriptive analysis, I consider it insufficient. The authors could make a difference by gender and by area.
  • I suggest dividing the "Resuls" section into three parts. 1. Descriptive analysis; 2. Regression analysis; 3. Differential analysis. For a better understanding of the reader.
  • It appears that there are differences by gender, but what analysis have the authors done? It is not clear.

We redid all the analysis.. as you can see.

Discussion

- The discussion is poor, and does not relate the results to enough recent studies.

We completed the informations…with other studies Lines 243-320

- There are no practical applications of the study or limitations. What was intended with this study? What does it respond to? What practical implications does this study have? What can be improved for subsequent similar studies?

As for the limits of the study, we mention the following: the study was conducted on a limited sample of students in Romania. It has included students from just one region in Romania. It is necessary to carry out such studies on a significant statistical sample at national level.

Although it is an interesting study, I consider that the authors should further exploit the data obtained, and restructure introduction and discussion, focusing more on the study objective. Very important to describe the importance of the study and practical implications.

We tried to make all the modifications.

Submission Date

25 April 2020

Date of this review

27 Apr 2020 16:58:08

Bottom of Form

© 1996-2020 MDPI (Basel, Switzerland) unless otherwise stated

Reviewer 2 Report

Manuscript ID: behavsci-799457

The manuscript is relevant and advances knowledge to merit publication, but I would like to suggest some major issues:

ABSTRACT

The aims of the study should be shown in the abstract. In addition, the age of the participants must also be shown (and, if possible, the mean and standard deviation) should also appear. In the abstract p must be in italic.

INTRODUCTION

The abbreviation (FMS) should appear behind fundamental motor skills.

The sentence “In choosing students for performance sports, coaches must start by learning certain factors represented by physical development and by the motor skills of students” is relatively true, as the literature has shown that coaches should also pay attention to psychological factors and cognitive determinants such as perception and decision making. For that reason, although the study focuses on fundamental motor skills, this phrase must be nuanced and name the other factors to be taken into account by sports coaches.

MATERIALS AND METHODS

Materials and methods section must be divided in subsections following: participants, instrument, procedure and statistical analysis.

Ethics statement must be in procedure subsection.

RESULTS

The statistical symbols should be in italics (p, χ²).

DISCUSSION

Again, although the authors have focused their study only on motor skills, in the discussion the authors should talk about the other factors that can influence sports performance (cognitive, decision making).

On the other hand, authors should recognize the limitations of their study. In this sense, they should consider that other variables influence sports performance, besides those related to skill execution, such as the mechanisms of perception and decision making. This can be the main limitation. In this sense, when it comes to selecting players for sports practice, is related to the measurement of motor skills outside the real context of the game, since there is a lack of elements that bias the extrapolation of the results obtained.

In addition, it would be useful to indicate future research derived from research carried out.

CONCLUSIONS

The authors should highlight the practical applications of their study.

REFERENCES

Review the author guidelines about references (e. g., abbreviation of journal names, years of publication of papers in bold type. . .).

Author Response

Top of Form

Open Review

English language and style

( ) Extensive editing of English language and style required 
( ) Moderate English changes required 
( ) English language and style are fine/minor spell check required 
(x) I don't feel qualified to judge about the English language and style 

Yes

Can be improved

Must be improved

Not applicable

Does the introduction provide sufficient background and include all relevant references?

( )

( )

(x)

( )

Is the research design appropriate?

( )

(x)

( )

( )

Are the methods adequately described?

( )

( )

(x)

( )

Are the results clearly presented?

( )

(x)

( )

( )

Are the conclusions supported by the results?

( )

(x)

( )

( )

Comments and Suggestions for Authors

Manuscript ID: behavsci-799457

The manuscript is relevant and advances knowledge to merit publication, but I would like to suggest some major issues:

ABSTRACT

The aims of the study should be shown in the abstract. In addition, the age of the participants must also be shown (and, if possible, the mean and standard deviation) should also appear. In the abstract p must be in italic.

We rethinked the abstract. As you can see.

INTRODUCTION

The abbreviation (FMS) should appear behind fundamental motor skills.

We made the modification. Line 73

The sentence “In choosing students for performance sports, coaches must start by learning certain factors represented by physical development and by the motor skills of students” is relatively true, as the literature has shown that coaches should also pay attention to psychological factors and cognitive determinants such as perception and decision making. For that reason, although the study focuses on fundamental motor skills, this phrase must be nuanced and name the other factors to be taken into account by sports coaches.

We made the modifications.. Thanks for outlining. Lines 40-43

MATERIALS AND METHODS

Materials and methods section must be divided in subsections following: participants, instrument, procedure and statistical analysis.

 We  made the modifications. Line 87-114

Ethics statement must be in procedure subsection.

 We moved the sentence

RESULTS

The statistical symbols should be in italics (pχ²).

We made the modifications

DISCUSSION

Again, although the authors have focused their study only on motor skills, in the discussion the authors should talk about the other factors that can influence sports performance (cognitive, decision making).

We made the modifications.. Lines 40-43

On the other hand, authors should recognize the limitations of their study. In this sense, they should consider that other variables influence sports performance, besides those related to skill execution, such as the mechanisms of perception and decision making. This can be the main limitation. In this sense, when it comes to selecting players for sports practice, is related to the measurement of motor skills outside the real context of the game, since there is a lack of elements that bias the extrapolation of the results obtained.

At the end of the article.

Study Limitations

As for the limits of the study, we mention the following: the study was conducted on a limited sample of students in Romania. It has included students from just one region in Romania. It is necessary to carry out such studies on a significant statistical sample at national level.

 In addition, it would be useful to indicate future research derived from research carried out.

 Study Limitations

As for the limits of the study, we mention the following: the study was conducted on a limited sample of students in Romania. It has included students from just one region in Romania. It is necessary to carry out such studies on a significant statistical sample at national level.

CONCLUSIONS

The authors should highlight the practical applications of their study.

 We added conclusions from line 323-341

REFERENCES

Review the author guidelines about references (e. g., abbreviation of journal names, years of publication of papers in bold type. . .).

We don’t know how to abbreviate all the journal names… so for the ones that we did not know the abreviations..we let it full

You can see the changes made in red

Submission Date

25 April 2020

Date of this review

29 Apr 2020 11:14:23

Bottom of Form

© 1996-2020 MDPI (Basel, Switzerland) unless otherwise stated

Round 2

Reviewer 1 Report

Although the authors have made some improvements, I consider that the article does not yet have the necessary scientific quality for publication. I suggest the following modifications:

  • Lines 97-107: Authors must rewrite this paragraph. It is divided on the one hand Instruments and measurement variables, and on the other hand Procedure. Regarding the procedure, it is not specified. When were the measures taken? Were they all taken at the same time? Who did you contact to carry out the analyzes?
  • Remove the reference to Anova on line 98. There should be another section within the method entitled Statistical analysis, which includes the name of the analyzes that have been carried out as well as the statistical program used.
  • Anova is a type of statistical analysis, not a program, as the authors respond. Was excel used? Spss? Specify. It should also be included that descriptive and correlation analyzes were made. Was the normality test performed?
  • The authors detail that correlation analyzes were performed but do not appear in the results. Also, they don't answer my question. Why wasn't a more demanding regression analysis performed?
  • Lines 343-346: It is not necessary to put a specific section for the limitations of the study. They are included in the discussion. The authors should detail the limitations of the study in greater depth, as I consider that they are insufficient.
  • In the discussion, the authors limit themselves to describing the differences found between the cities, but why do the authors believe that there are these differences between the cities? The discussion should be more detailed, supported by other previous studies.
  • I suggest the authors to read other related articles from this magazine, to improve the article format.

Author Response

First of all. We made the second round of modifications in purple to be more simple to identify them. The first round of modifications were made in red.

  • Lines 97-107: Authors must rewrite this paragraph. It is divided on the one hand Instruments and measurement variables, and on the other hand Procedure. Regarding the procedure, it is not specified. When were the measures taken? Were they all taken at the same time? Who did you contact to carry out the analyzes?
  • With all of the modifications now you can find the information from line 83-126
  • Materials and Methods
  • 1. Aim of the study
  • In the study we appraised bodyweight and height among the students in the research sample and used the results obtained by the students in motor skills tests (starting from the national standards). The results obtained have been calculated per county and by gender in order to make the correlation between the value of the Body Mass Index and the motor skills of students.
  •  
  • 2 Participants
  • The research study was conducted on a sample of 220 fifth graders from the rural areas of the Vrancea County (61 students), Iasi (72 young people) and Suceava (87 adolescents) within the Moldavian area. Repartition by gender is inequal, as we have examined 116 boys and 104 girls. In each county, we included within the study the students within two communes, given that the number of students has been on a decreasing trend in the rural areas. The students appraised are aged between 10 and 12 years old.
  •  
  • 3. Procedure
  • Subsequently, we made the correlation between the value of Body Mass Index and the motor skills assessed using the for aforementioned tests. The tests were applied throughout January 2020 to all students, during physical education and sports classes. In order to carry out the tests, we collaborated with the physical education teachers from the respective schools (the performance of these tests is required at the national level).
  • Among these young people, we have assessed the value of bodyweight and the motor skills. The value of bodyweight was appraised using the Body Mass Index. The formula is the same as for the adults, but result interpretation is different by age group and gender [18]. The values comprised between 11 and 84% are normal; those exceeding 84% indicate overweight or obesity, while values under 11% indicate underweight.
  • The motor skills were assessed through 4 tests included in the grade national standard [17] for the fifth grade: push-ups; standing softball throw; standing long jump; upper body extension, face down. The grade standard is different by gender. To avoid a too great dispersion of the results, their appraisal starts from the grades 5 and 6 – 7,8 and 9 – 10 (Table 1).
  • Table 1 – National grade standard in the fifth grade

Vote

Fifth grade

Under 5

5-6

7-9

10

Push-ups (rep.) M

Under 4

4-6

8-12

14

Push-ups (rep.) F

Under 2

2-3

4-6

7

Standing softball throw (m) M

Under 18

18-19

20-22

23

Standing softball throw (m) F

Under 12

12-13

14-16

17

Standing long jump (cm.) M

Under 150

150-153

156-162

165

Standing long jump (cm.) F

Under 130

130-133

136-142

145

Face down, upper body extension (rep.) M

Under 16

16-17

19-23

25

Face down, upper body extension (rep.) F

Under 15

15-16

17-21

23

  •  
  • 3 Statistical analysis
  • Collected data were subjected to statistical computation, using the ANOVA one-way algorithm included in MsExcel, to calculate the descriptive statistics (mean, standard error) and find out whether there were significant differences and upgraded with PostHoc Daniel's XL Toolbox version 4.01 (http://xltoolbox.sf.net), to identify the differences.
  •  
  • 4 Ethics Statement
  • All subjects gave their informed consent for inclusion before they participated in the study. The study was conducted in accordance with the Declaration of Helsinki, and the protocol was approved by the Ethics Committee of the Faculty of Physical Education and Sport from “Alexandru Ioan Cuza” University of Iasi, Romania.
  • Remove the reference to Anova on line 98. There should be another section within the method entitled Statistical analysis, which includes the name of the analyzes that have been carried out as well as the statistical program used.
  • We mentioned the method entitled for statistical analisys and included all the details in a special section. Lines 116-120
  • Collected data were subjected to statistical computation, using the ANOVA one-way algorithm included in MsExcel, to calculate the descriptive statistics (mean, standard error) and find out whether there were significant differences and upgraded with PostHoc Daniel's XL Toolbox version 4.01 (http://xltoolbox.sf.net), to identify the differences. Lines
  •  
  • Anova is a type of statistical analysis, not a program, as the authors respond. Was excel used? Spss? Specify. It should also be included that descriptive and correlation analyzes were made.
  •  

As we said earlier we made all the modifications. Lines 116-120

  • Collected data were subjected to statistical computation, using the ANOVA one-way algorithm included in MsExcel, to calculate the descriptive statistics (mean, standard error) and find out whether there were significant differences and upgraded with PostHoc Daniel's XL Toolbox version 4.01 (http://xltoolbox.sf.net), to identify the differences.
  • ANOVA within rows, between groups for different superscripts, one by one comparison:
  • ns: not significant; * significant, (P <0.05); distinguished significant = ** (P <0.01);  highly significant =  *** (P <0.001)

  • Was the normality test performed?

We didn’t use the normality test because we used the data collected to calculate the BMI for the students included in the study. The BMI was calculated on a online platform in which you include the age, gender, height and weight and when you press calculate, they tell you the result and the interpretation of the result (underweight, normal weight and overweight). We didn’t need to perform a normality test, the platform that we used already calculated the normality vs underweight and overweight..

  • The authors detail that correlation analyzes were performed but do not appear in the results. Also, they don't answer my question. Why wasn't a more demanding regression analysis performed?

We didn’t performed a regression analysis because you can see how can the tests results correlated with the BMI, answered our questions. We did a ANOVA within rows, between groups for different superscripts, one by one comparison: ns: not significant; * significant, (P <0.05); distinguished significant =  ** (P <0.01);  highly significant =  *** (P <0.001).

  • Lines 343-346: It is not necessary to put a specific section for the limitations of the study. They are included in the discussion. The authors should detail the limitations of the study in greater depth, as I consider that they are insufficient.

. We moved the paragraph right after discussions. Lines 346-350

  • As for the limits of the study, we mention the following: the study was conducted on a limited sample of students in Romania. It has included students from just one region in Romania. It is necessary to carry out such studies on a significant statistical sample at national level. We also specify that is very important to take in consideration the mental and cognitive characteristics of the students if they will be oriented towards sports performance.

  • In the discussion, the authors limit themselves to describing the differences found between the cities, but why do the authors believe that there are these differences between the cities? The discussion should be more detailed, supported by other previous studies.
  • We inserted the motivation in choosing the cities and new references that confirm the choosing of the cities. Lines 256-267
  • Numerous research shows that the environment can influence the physical activity of the individual and indirectly his health [19]. Populations living at high altitudes have a slight delay in linear growth and a larger diameter and chest circumference of the inhabitants of regions near the sea level. These differences are attributed to the phenomenon of hypobaria, as well as socioeconomic, nutritional and environmental [20].
  • Human body prioritises which segments to grow when there is limited energy available for growth, such as at high altitude. This comes at the expense of other segments, for example the lower arm. The body may prioritise full growth of the hand because it is essential for manual dexterity, whilst the length of the upper arm is particularly important for strength [21].
  • The present study analyzed the results obtained by students working in schools located on different altitudes (200-250 m altitude for Iasi, 300-350 m altitude for Vrancea, 500-550 m altitude for Suceava).

  • I suggest the authors to read other related articles from this magazine, to improve the article format.
  • Thank you for the suggestions. We made the modifications.

Best regards, 

Raluca Onose

Reviewer 2 Report

Manuscript ID: behavsci-799457

The manuscript is relevant and advances knowledge to merit publication, but I would like to suggest some issues:

ABSTRACT

The aims of the study should be shown in the abstract. Some databases only index the abstract and keywords, and readers and researchers read the abstract first. It is impossible to know the scope of an investigation without knowing its objectives. So, it is important that the objective is shown in the abstract.

MATERIALS AND METHODS

Authors should indicate the statistical program used.

CONCLUSIONS

Authors should recognize the limitations of their study. In this sense, they should consider that other variables influence sports performance, besides those related to skill execution, such as the mechanisms of perception and decision making. For me, this is the main limitation of the study conducted by the authors. In this sense, when it comes to selecting players for sports practice, is related to the measurement of motor skills outside the real context of the game, since there is a lack of elements that bias the extrapolation of the results obtained.

Author Response

Manuscript ID: behavsci-799457

The manuscript is relevant and advances knowledge to merit publication, but I would like to suggest some issues:

 First of all. We made the second round of modifications in purple to be more simple to identify them. The first round of modifications were made in red.

ABSTRACT

The aims of the study should be shown in the abstract. Some databases only index the abstract and keywords, and readers and researchers read the abstract first. It is impossible to know the scope of an investigation without knowing its objectives. So, it is important that the objective is shown in the abstract.

We inserted the aim of the study in the abstract. You can find the modification at lines 10-11.

The aim of the study: the evaluation of the students' motor skills in order to orient towards certain sports branches. 

MATERIALS AND METHODS

Authors should indicate the statistical program used.

We inserted more details regarding the program that we used at lines 116-120.

Collected data were subjected to statistical computation, using the ANOVA one-way algorithm included in MsExcel, to calculate the descriptive statistics (mean, standard error) and find out whether there were significant differences and upgraded with PostHoc Daniel's XL Toolbox version 4.01 (http://xltoolbox.sf.net), to identify the differences.

CONCLUSION

Authors should recognize the limitations of their study. In this sense, they should consider that other variables influence sports performance, besides those related to skill execution, such as the mechanisms of perception and decision making. For me, this is the main limitation of the study conducted by the authors. In this sense, when it comes to selecting players for sports practice, is related to the measurement of motor skills outside the real context of the game, since there is a lack of elements that bias the extrapolation of the results obtained.

We made the modifications.

Motor skills are just one of the essential elements for orientation to a particular sport. It is also important to evaluate physical development along with knowing the mental characteristics, especially those related to perception and decision making. Lines 3346-350 and 371-375

Round 3

Reviewer 1 Report

The authors have made an improvement to the article.
From my point of view they should carry out a regression analysis,
since it would give more scientific quality to the paper.

Author Response

We made the modifications according to the suggestions from the reviewer 1. The regression analysis was carried for the scientific quality of the paper.

From line 122-141

2.3 Statistical analysis

The statistical analysis applied in this paper aimed at identifying and quantifying the links between body mass index (BMI) and a number of potentially explanatory, quantitative variables (number of push-ups, softball throwing in m, standing long jump in m and number of upper body extensions). The statistical calculations as well as the graphical representations were made using Excel 2016.

For this purpose, the linear regression method was applied, which is a statistical method that allows the identification and quantification of the links caused between one (single regression) or more (multiple regression) quantitative explanatory variables (predictors) and a dependent variable (response), expressed, also quantitatively.

Regarding the values of the coefficient, they can oscillate between -1 and 1, indicating the intensity of the relationship between the variables; therefore, the closer the value is to -1 or 1, the closer the relationship.

Negative values indicate inverse proportionality relationships while positive values indicate direct proportionality relationship. Raised to square, the correlation coefficient becomes the coefficient of determination (R2), expressing the fraction of the variance of the dependent variable explained by the explanatory variables.

Also, the validation of the regression model involves testing the statistical significance of the correction coefficient, the free coefficient and the partial regression coefficient. For this purpose, a series of statistical tests are used, for which the probability of error must be less than 5% (p˂0.05).

From line 170-263

The  regression analysis for this study.

Table 5 highlights the results on the correlation between body mass index (BMI) and the motor skills tests.

Table 5 - The correlation between BMI and the motor tests applied by regions

MST

Sex

Iasi

Suceava

Vrancea

n

BMI

MTR

p-value

n

BMI

MTR

p-value

n

BMI

MTR

p-value

Push-ups

M

42

17.68±0.53

11.05±1.00

2.27 x 10-22

***

43

18.42±0.45

7.98±0.89

2.24x10-26

***

31

18.32±0.41

9.97±0.48

1.56x10-11

***

F

30

16.55±0.58

9.93±0.97

8.99 x 10-15

***

44

18.06±0.48

4.18±0.46

8.45x10-26

***

30

16.07±0.42

7.70±0.33

9.39x10-10

***

Softball throw

M

42

17.68±0.53

27.27±0.83

4,23x10-08

***

43

18.48±0.45

24,12±0,93

4,20x10-14

***

31

18.32±0.41

21,23±0,57

9,73x10-08

***

F

30

16.55±0.58

18.02±0.28

0,039

***

44

18.06±0.48

17,73±0,62

2,81x10-11

***

30

16.07±0.42

15,67±0,46

0,0001

***

Standing long jump

M

42

17.68±0.53

1.37±0.05

2,22x10-10

***

43

18.48±0.45

1,41±0,03

2,83x10-09

***

31

18.32±0.41

1,42±0,04

4,84x10-08

***

F

30

16.55±0.58

1.34±0.04

1,04x10-05

***

44

18.06±0.48

1,40±0,03

4,79x10-12

***

30

16.07±0.42

1,33±0,03

8,12x10-05

***

Upper body extensions

M

42

17.68±0.53

33.17±2.03

3,64x10-14

***

43

18.48±0.45

25,37±0,74

2,16x10-13

***

31

18.32±0.41

30,94±1,74

5,44x10-14

***

F

30

16.55±0.58

31.90±1.62

9,71x10-07

***

44

18.06±0.48

23,36±0,69

2,11x10-12

***

30

16.07±0.42

24,23±1,22

2,71x10-09

***

MST- Motor skills tests; MTR=Mean of test results (rep./m); M=male; F=female;  = mean and standard deviation; *** p˂0,001 = very significant.

As Table 5 indicates, the average value for BMI of the boys in the Iasi region was 17.68 ± 0.53, and the average in terms of the number of push-ups was 11.05 ± 1.00 reps.

Figure 1.a shows the relationship between BMI and the number of push-ups performed by boys, and a positive regression can be observed (F6,00˃Fsig. 0.018), therefore, the ability to perform push-ups is influenced in 53% by the BMI (Figure 1.a).

With regard to the male subjects from the Suceava region, the average value for BMI was 18.48 ± 0.45 and the average number of push-ups was 7.98 ± 0.89 repetitions (Table 5). Regarding the influence of the ability to perform push-ups by BMI, the value of the coefficient of determination (R2) is 5x10-6, the regression being also positive (F0.88˃Fsig. 0.35) (Figure 1.b).                                                     

For the Vrancea region, the average values obtained in terms of BMI were 18.32 ± 0.41, while for the number of push-ups the average was at a level of 9.97 ± 0.48 reps. In this case, a positive regression was recorded (F0.59˃Fsig. 0.44), therefore the ability to perform push-ups is influenced in 19% of the cases by the BMI (Figure 1.c).

From a statistical point of view, significant differences were highlighted, the values of p -value being higher than 0.001 (Table 5).

1.a-Iasi

1.b-Suceava

1.c-Vrancea

Figure 1 - Relationship between the BMI and the number of push-ups performed by the male subjects in the three regions

With regard to the female subjects from the three analyzed regions we can observe, similarly as in the case of the male subjects, that there is a positive regression. Therefore, for girls in the Iasi region, the mean BMI was 16.55 ± 0.58 and the number of push-ups was 9.39 ± 0.97 reps (Table 5). Regarding the significance of the differences between the averages, the difference is very significant (p-value = 8.99 x 10-15) this being lower than the value 0.001. Regression analysis F2.21˃Fsig. 0.14 shows a positive regression, therefore, the ability to perform push-ups is influenced in 17% of the cases by the BMI (Figure 2.a).

2.a-Iasi

2.b-Suceava

2.c-Vrancea

Figure 2 - Relationship between the BMI and the number of push-ups performed by the girls in the three regions

The analysis of BMI data correlated with the number of push-ups performed by the girls in the Suceava region shows a positive regression (F2.82˃Fsig. 0.10), the ability to perform push-ups being influenced by only 0.1% (Figure 2.b)

For the girls from the Vrancea region the average BMI was 16.07 ± 0.42 and for the number of push-ups an average of 7.70 ± 0.33 reps was obtained; p-value was 9.93x10-10 the difference being also very significant (p-value˃0.001) (Table 5). The ability to made push-ups is influenced in 15% girls by the BMI (R2 = 0.15) (Figure 2.c).

Another test analyzed was the softball throw, the average values obtained for both boys and girls being also correlated with the BMI.

Therefore for the boys from the Iași region the average value obtained for this exercise was 27.27 ± 0.83 m, the coefficient of determination (R2) being equal to 1 (Figure 3.a) the regression being a negative one (F0,18˃Fsig. 0 , 66), therefore, the BMI does not influence the results obtained in this test.

The boys from the Suceava region recorded an average of 24.12 ± 0.93 m (Table 5) the correlation in this case being a positive one (F1.48˃Fsig. 0.22) the softball throwing test being influenced by the BMI (R2 = 0.035) (Figure 3.b).

For the boys from the Vrancea region, as well as for those from Iași, the regression was negative (F0.37˃Fsig. 0.54), the average for this sample being 21.23 ± 0.57 m. (Table 5). The difference regarding the p-value (9.73x10-08) is highly significant.

3.a-Iasi

3.b-Suceava

3.c-Vrancea

Figure 3 - Relationship between the BMI and the softball throwing test for the male subjects in the three regions

For girls, the correlation between the softball throwing test and BMI showed a negative regression for the group from the Suceava region (F0.54˃Fsig. 0.46), R2 recording a value of 0.0127 (Figure 4.b) and for the one from the Vrancea region where the value of R2 was equal to 1 (Figure  4.c), F3.89˃Fsig.0.058. For the girls from the Iași region, the regression was positive, R2 = 0.0012 (Figure 4.a). With regard to the p-value, the data displayed in Table 5 indicates that there the differences are highly significant.     

4.a-Iasi

4.b-Suceava

4.c-Vrancea

Figure 4 - Relationship between the BMI and the softball throwing test for the female subjects in the three regions

Regarding the standing long jump correlated with BMI for the groups of boys from the Iași and Suceava regions, the regression was positive F1.13˃Fsig. 0.29 for the Iași group and F4.45˃Fsig. 0.041 for the Suceava group. Negative regression F0.078˂Fsig 0.781 was recorded in boys from the Vrancea region. (Figure 5.abc).

5.a-Iasi

5.b-Suceava

5.c-Vrancea

Figure 5 - Relationship between the BMI and the standing long jump test for the boys in the three regions

For the girls from the three analyzed regions, the averages obtained for softball throwing test were 1.34 ± 0.44m for the Iași region, 1.40 ± 0.03m for the Suceava region and 1.33 ± 0.03m for the Vrancea region (Table 5).

The regression for the groups of girls from Iași and Suceava was positive F1.69˃Fsig. 0.20 and F18.02˃Fsig. 0.0001 (Figure 6.a,b). The values obtained for thr p-value were 1.03x10-05 for the girls from the Iași region and 4.79x10-12 for those from the Suceava region.

For the group of girls from the Vrancea region, a negative regression F0.15˂Fsig 0.69 was obtained, the differences being also quite significant, p-value = 8.12x10-05. The values obtained for the coefficient of determination (R2) were 1 (Iasi), 0.30 (Suceava) and respectively 1 (Vrancea) (Figure 6.abc).

6.a-Iasi

6.b-Suceava

6.c-Vrancea

Figure 6 - Relationship between the BMI and the standing long jump test for the female subjects in the three regions

For the Trunk Extensions Test, the boys from the Iași region recorded an average of 33.17 ± 2.03 reps (Table 5). The correlation between BMI and this type of test, in this case, was a positive one F0,65˃Fsig. 0.42, R2 = 1 (Figure 7.a).

For those in the Suceava region, the recorded average was 25.37 ± 0.74 reps and the correlation between BMI and this test was also positive F6.60˃Fsig. 0.01, R2 = 1 (Figure 7.b).

The last group analyzed, the one in the Vrancea region, recorded an average of 30.94 ± 1.74 and the correlation between BMI and this test was negative F0.23˂Fsig. 0.63 with an R2 = 1 (Figure 7.c).

7.a-Iasi

7.b-Suceava

7.c-Vrancea

Figure 7 - Relationship between the BMI and the upper body extension test for the boys in the three regions

Regarding the female group from the Iasi region, the recorded average was 31.90 ± 1.62 reps (Table 5). The analysis of the correlation between BMI and the upper body extension test was positive F1.99˃Fsig. 0.17, R2 = 1 (Figure 8.a).

The group of girls from the Suceava region recorded an average of 23.36 ± 0.69 reps and the correlation between BMI and this test is F5.68˃Fsig. 0.002, the regression being positive, with R2=1 (Figure 8.b).

The representative female group from the Vrancea region had an average of 24.23 ± 1.22 reps and the regression analysis showed a positive correlation with F6.96˃Fsig. 0.001 with R2=1      (Figure 8. c).

8.a-Iasi

8.b-Suceava

8.c-Vrancea

Figure 8 - Relationship between the BMI and the upper body extension test for the female subjects in the three regions

From line 380-397 Conclusions

The BMI push-ups test showed a positive regression for all the groups under analysis, both in the tests performed by boys and the ones performed by girls.

In the softball throwing test for boys, the regression was negative in the counties of Iasi and Vrancea (there was no correlation between BMI and the softball throwing test) as opposed to the positive regression recorded in Suceava.

For the group of girls from the Suceava and Vrancea counties, the correlation between BMI and softball throwing test was negative, whereas for the girls from Iasi, the correlation between BMI and softball throwing was positive.

In the standing long jump test for the groups of boys from Iasi and Suceava, the regression analysis was positive, while the analysis of the correlation between BMI and this test in boys from the Vrancea region, regression was negative. In the groups of girls from the Iasi and Suceava regions, the comparison between BMI and the applied test showed a positive regression, while in the girls from Vrancea, the resulting regression was negative.

In the trunk extension test, the regression for the boys was positive in the Iasi and Suceava regions and negative in the Vrancea region compared to the groups of girls who, when analyzing the correlation between BMI and the applied test, showed a positive result.

The differences obtained according to p-value showed very significant differences for all motor skills tests and for all study groups.
